# Efficacy of a Mobile App-Based Coaching Program for Addiction Prevention among Apprentices: A Cluster-Randomized Controlled Trial

**DOI:** 10.3390/ijerph192315730

**Published:** 2022-11-25

**Authors:** Severin Haug, Nikolaos Boumparis, Andreas Wenger, Michael Patrick Schaub, Raquel Paz Castro

**Affiliations:** 1Swiss Research Institute for Public Health and Addiction, Zurich University, Konradstrasse 32, 8005 Zurich, Switzerland; 2Marie Meierhofer Children’s Institute, Pfingstweidstrasse 16, 8005 Zurich, Switzerland

**Keywords:** substance use, addiction, prevention, apprentices, adolescents, mobile app

## Abstract

Background: Addictive behaviors such as tobacco/e-cigarette smoking, at-risk alcohol consumption, cannabis use or compulsive internet use are common among apprentices. *ready4life* is a mobile app-based preventive intervention program for apprentices that promotes life skills and reduces risky behavior. The present study tested the efficacy of *ready4life* for addiction prevention among apprentices in Switzerland within a controlled trial. Methods: Two-arm, cluster-randomized controlled trial including assessments at baseline and follow-up after 6 months. Participants of the intervention group received coaching by a conversational agent for 16 weeks. The main outcome measure was a composite score for addictive behaviors, which included (1) at-risk drinking, (2) tobacco/e-cigarette smoking, (3) cannabis use and (4) problematic internet use. Results: A total of 2275 students from 159 vocational school classes in Switzerland, were invited for study participation. Of these, 1351 (59.4%) students with a mean age of 17.3 years and a male proportion of 56.6% provided informed consent to participate. The follow up assessment at month 6 was completed by 962 (71.2%) study participants. The results concerning the primary outcome showed a stronger decrease of addictive behaviors between baseline and follow up in the intervention group compared to the control group. In particular, significant effects were observed for at-risk drinking and problematic Internet use, while no significant effects were observed for tobacco/e-cigarette smoking and cannabis use. Conclusions: The majority of apprentices invited for program participation within vocational schools participated in the *ready4life* program for addiction prevention. The mobile app-based coaching was effective in reducing risk behaviors such as at-risk drinking and problematic Internet use in a group of adolescents who have an especially high risk of engaging in addictive activities.

## 1. Introduction

For young people, entering professional life is associated with numerous changes and challenges. Lower financial dependency and detachment from parents increase the degree of autonomy, but at the same time the responsibility for one’s own actions increases. Everyday working life is often associated with stress, time and success pressure, and dealing with coworkers, superiors and clients poses new demands on the social skills of trainees. Consequently, the start of the apprenticeship is linked to potential health risks, especially increased substance use. A representative study among adolescents from Switzerland aged 15 to 19 years showed a significantly higher proportion of regular (at least monthly) tobacco smokers (21% vs. 12%) and alcohol users (60% vs. 52%) among adolescents in vocational than general education, whereas there were no significant differences concerning regular cannabis use (12% vs. 8%) [1]. Apprentices, who are mostly in late adolescence, are a heterogeneous group in terms of risk behaviors, with the predominant share showing at least one or co-occurring addictive behaviors [2].

Life-skills training programs addressing social skills and influences are particularly suitable for addiction prevention in younger adolescents and those who have not yet started using addictive substances [3]. In later adolescence, interventions based on motivational and cognitive-behavioral principles are promising but the published findings are mixed and the effects sizes typically small [3,4]. Due to the age and heterogeneous risk profiles among apprentices, interventions including both general life-skills training modules and motivational, cognitive-behavioral modules to reduce specific substances, show potential.

Schools provide an ideal access route to a large number of young people. However, large-scale implementation and dissemination of traditional coaching programs, which are provided face-to-face also pose serious challenges [5]. On the one hand, teachers or other professionals need the time, motivation and skills, on the other hand, resources in terms of staff, money and time are necessary to implement these programs.

Digital interventions have great potential to overcome these implementation obstacles. They can achieve broad reach at low cost and provide the possibility of delivering individualized content automatically. Moreover, they might be more appealing as they allow anonymous participation and tailoring contents to personal needs [6,7].

A review of digital interventions targeting multiple lifestyle risk behaviors in the context of schools found short-term benefits for programs focusing on diets, physical activity, and screen time, while no effects were observed for alcohol or tobacco use [8]. The studies’ lack of examination of mobile health interventions and the paucity of data on effectiveness beyond the immediate effect of the intervention are highlighted by the authors as shortcomings. A recent review of app-based interventions to prevent substance use among adolescents and adults revealed some evidence of effectiveness [9]. Among the 17 articles analyzed, most interventions focused on alcohol reduction. While the non-randomized studies tended to provide positive results regarding effectiveness, the results of the randomized controlled trials were mixed. The authors concluded that further studies are required with adequate sample sizes and longer follow-up assessments.

Mobile phone-based interventions offer a promising way to deliver preventative services. Similar to other developed nations, almost all (99%) adolescents between the ages of 12 and 19 in Switzerland own a mobile phone [10]. The potential effectiveness of text messaging-based programs for lowering alcohol and cigarette use among at-risk groups, including young people, has been highlighted in recent reviews [7,11,12].

Participation in a web- and text messaging-based coaching program resulted in a reduction of binge drinking among vocational and upper secondary school students in Switzerland [13]. Another Swiss study showed the acceptance and potential effectiveness of a previous version of *ready4life*, a text messaging-based life-skills training program for the prevention of substance use among vocational school students [14]. The individualized coaching program addressed self-management-, social-, and substance use resistance-skills. Program use was stimulated through interactive features such as quiz questions, contests, and a friendly competition. For the program, four out of five vocational students could be reached in school classes and pre-post comparisons revealed decreased perceived stress and at-risk alcohol use between baseline and a follow-up assessment at month 6 [14].

Within the current study, we tested the efficacy of the current *ready4life* version, a mobile app-based intervention program for addiction prevention in apprentices. This study provides the first randomized controlled trial on a mobile app-based addiction prevention program among apprentices. The program addresses both apprentice‘s risks and resources by promoting life skills (coping with stress, social skills) on the one hand and reducing various risk behaviors (problematic Internet use, tobacco/e-cigarette smoking, cannabis use, at-risk drinking) on the other. We hypothesized that participating in the 4-month mobile app-based coaching program would be more effective than assessment only for reducing addictive behaviors including (1) at risk-drinking, (2) tobacco/e-cigarette smoking, (3) cannabis use, and (4) problematic Internet use at 6 months follow-up.

## 2. Methods

### 2.1. Study Design

This study aimed at testing the efficacy of *ready4life*, a mobile app-based coaching program for addiction prevention among apprentices. A two-arm, parallel group, cluster-randomized controlled trial was conducted, including assessments at baseline and follow-up at month 6. The effects of the intervention was contrasted with an assessment-only control group.

### 2.2. Participants, Setting and Procedure

About half of all Swiss adolescents ages 16 to 19 attend vocational schools [15], with teens aged 17 (males: 60%, females: 47%) and 18 (males: 57%, females: 45%) having the highest proportions. Within the present study, vocational schools in Switzerland’s German speaking region were invited to participate by prevention experts from collaborating regional centers for addiction prevention. The experts were trained and informed on recruitment for the *ready4life* intervention program. They then scheduled sessions during regular school lessons designated for health education in the participating vocational school classes. During these sessions, the students were informed about the program and the accompanying study, including its aims, assessments, reimbursement, and data protection. The chatbot coaching program was presented, using a teaser video (www.r4l.swiss, (accessed on 23 November 2022)). In order to increase the response rate for the follow-up assessment survey and thus representativeness of the sample [16], students were offered a reward for participating in the follow-up assessment (cash 10 CHF (US $10.50)). Students were invited to install the app on their smartphone and to complete online study registration and baseline assessment. After providing informed consent, they were asked to select a username and enter their mobile phone number.

Participants from school classes that belonged to the intervention group received an individualized profile on their risks and resources (see also intervention program) and could select two out of the following six coaching topics: stress, social skills, social media & gaming, tobacco/e-cigarette smoking, cannabis, and alcohol. Subsequently, they received coaching by the program, for a period of four months (two months per topic) on their two chosen topics. Study participants in control classes did not participate in the intervention program but were invited for program participation after completed follow-up assessment in month 6. Participants of both study groups were invited to complete the online follow-up assessment via SMS text messaging. Non-responders were additionally contacted by research assistants and invited to participate in a phone-based follow up survey (computer assisted telephone interview).

### 2.3. Randomization and Allocation Concealment

A cluster-randomized controlled trial utilizing “school class” as a randomization unit was carried out to prevent spillover effects within school classes. We used a separate randomization list for each school (stratified randomization) and employed block randomization with computer generated randomly permuted blocks of 4 cases to approximate equality of sample sizes in the study groups [17].

The recruiting prevention experts were blinded to the group allocation of school classes until the study participants had provided their informed consent, username, mobile phone number, and baseline data. Additionally, the research assistants performing the computer-assisted follow up assessments were blinded of the group assignment during assessments of the primary and secondary outcomes.

### 2.4. Sample Size Calculation

We expected a small effect for our primary outcome of this study, the composite measure of addictive behaviors, based on previous evaluations of digital programs for the prevention of addictive behaviors [13]. Based on a calculation with G-Power and an estimated Cohen’s d of 0.2, a sample size of *n* = 412 in each study group was needed to have 80% power for a Wilcoxon-Mann–Whitney-test (=5%, 2-sided) in order to detect this difference. Due to the nesting of vocational school students within school classes, we also considered a potential design when determining the sample size for our study. We anticipated an average cluster size of 13 participants per school class and an intra-cluster correlation coefficient of 0.05 based on similar studies in vocational schools [13,14]. Multiplying the resulting design effect of 1.60 by the required size for an un-nested sample (*n* = 412) yielded a minimum sample size of *n* = 659 per study group and a total of *n* = 1318 study participants.

### 2.5. Intervention Program

*ready4life* (www.r4l.swiss, (accessed on 23 November 2022)) is a mobile app-based program for addiction prevention by promoting life skills and reducing risk behaviors. It provides individualized coaching by a conversational agent. Details of the intervention program are presented in the study protocol of this trial [18].

The chatbot asked participants to select a male or female avatar, submit demographic information (age and sex), and complete baseline questionnaires on stress, self-efficacy, social skills, Internet use, tobacco/e-cigarette, cannabis use, and alcohol consumption. Based on this survey, an individual feedback was created, which highlighted areas where a participant had sufficient resources and where coaching might be necessary. Taking this feedback into account, participants could select two out of six possible program modules: (1) stress, (2) social skills, (3) social media & gaming, (4) tobacco/e-cigarette smoking, (5) cannabis, and (6) alcohol. Utilizing information from the baseline assessment for intervention tailoring, program participants received coaching for each of the two chosen topics for a total of 8 weeks. The virtual coach encouraged the participants to consume addictive substances responsibly, provided feedback on current use and life skills, and offered tailored information in weekly dialogues. An average weekly dialogue took between two and five minutes to process. The app’s “ask the exert” chat feature allowed users to ask addiction prevention experts specific or individual questions. Several interactive elements, including quizzes, challenges, and a playful competition were integrated into *ready4life* in order to encourage active program engagement. Users of the program could earn credits for each completed weekly conversation with the chatbot. Participants had a chance to win one of several alluring prizes in a prize draw when the program was over, and their chances of winning increased the more credits they had collected.

The framework and intervention components of each program module were comparable, and they were all based on the Social-Cognitive Theory (e.g., goal-setting, self-monitoring) [19], the Social Norms Approach (e.g., normative feedback) [20] and Motivational Interviewing (e.g., decisional balance) [21].

### 2.6. Assessments and Outcomes

Individual demographic information (age, sex) as well as characteristics of the schools and school classes were assessed at baseline. Both, baseline- and follow-up assessments included the subsequent addictive behaviors and life skills targeted in the prevention program:At risk-drinking in the preceding 30 days, as per guidelines from the Swiss Federal Office of Public Health [22]. At risk-drinking was present, if (1) the maximum consumption on one occasion in the preceding 30 days was higher than 4/5 (female/male) alcoholic standard drinks (10–12 g of pure alcohol) or (2) the total consumption in the preceding 30 days was higher than 20/40 (female/male) drinks or (3) the number of alcohol consumption days in the preceding 30 days was higher than 20.30 days point prevalence for tobacco/e-cigarette smoking, defined as having smoked within the past 30 days [23].Number of tobacco cigarettes smoked in the previous 30 days by multiplying the number of cigarettes smoked on a typical smoking day and the number of cigarette smoking days.Cannabis use days in the preceding 30 days.Problematic Internet use evaluated by the Short Compulsive Internet Use Scale (CIUS-5) with a cut-off of nine points [24].General self-efficacy measured by the Short Scale for Measuring General Self-efficacy Beliefs [25].Self-perceived stress evaluated by a single-item measure of stress symptoms [26].

The primary outcome was a composite measure (range: 0–4) for addictive behaviors, which reflects the number of risk behaviors (at-risk drinking in the preceding 30 days, 30 days point prevalence for tobacco/e-cigarette smoking, 30 days point prevalence for cannabis use, problematic Internet use) that a person exhibits. Secondary outcomes were dichotomous measures reflecting the single risk behaviors (1) at risk drinking in the preceding 30 days, (2) 30 days point prevalence for tobacco/e-cigarette smoking, (3) 30 days point prevalence for cannabis use, and (4) problematic Internet use as well as metric measures reflecting (5) total number of alcoholic drinks consumed in the preceding 30 days, (6) number of tobacco cigarettes smoked in the preceding 30 days, (7) number of cannabis use days in the preceding 30 days, (7) extent of problematic Internet use (CIUS-5 total score) [24], (9) general self-efficacy [25] and (10) self-perceived stress [26].

### 2.7. Statistical Analysis

By using chi-square tests for categorical data and t-tests for continuous variables, we looked at baseline differences between individuals in the two groups. To test whether participants lost to follow-up differed from the respondents, we performed (generalized) linear mixed models (GLMMs) while modeling a random intercept for school class.

GLMMs were used to assess intervention effects for binary outcomes and Linear Mixed Models (LMM) for continuous outcomes [27,28]. The primary outcome representing the composite measure for addictive behaviors (range: 0–4) was analyzed as a binomial variable (number of non-risks, number of risks) with a GLMM as described in [29].

Within each (G)LMM, a random intercept was modelled for school class. Analyses of binary and continuous outcomes included follow-up scores as the dependent variable. Independent variables included group as a predictor, as well as baseline values for each respective variable.

To identify potential moderators, (G)LMMs with a random intercept for school class were modelled. In addition, interaction terms for age, sex, the dichotomous baseline variables and study group were included, one at a time. All data were analyzed according to complete case analysis and the intention-to-treat principle (ITT). For ITT analyses, we used multiple imputation procedures [30] and performed imputations for each group separately to preserve within-group homogeneity.

Predictors of missing data at follow-up were sex, age, school class, and, particularly for the intervention group, program engagement. Binary outcomes were imputed using logistic regression and continuous outcomes using predictive mean matching. After examining 20 datasets, no systematic bias in convergence was revealed. The results from the imputed dataset were crosschecked with the non-imputed data. A type I error rate of *p* < *0*.05 on two-sided tests was considered statistically significant when evaluating intervention main effects and moderator effects. All analyses were performed using R, version 4.1.2. Multiple imputations were conducted with R’s mice package [30] and (G)LMMs with the lme4 package [31].

## 3. Results

### 3.1. Study Participants

Figure 1 depicts participants’ progression through the trial. At online screening assessment, 2275 students were present in 159 vocational school classes. Of these, 1351 (59.4%) provided informed consent and a mobile phone number, and therefore, participated in the study. A total of 688 students from 76 school classes were randomly assigned to the intervention group, 663 students from 77 classes were assigned to the control group. Primary outcome assessment at six-month follow-up was completed by 949 students (70.2%).

Baseline characteristics for the study sample are presented in Table 1. The mean age was 17.3 years (SD 3.0) and 43.4% of the participants were female.

There were no baseline differences between the intervention and control group.

Participants who dropped out reported a greater quantity of cigarettes smoked in the preceding 30 days (OR = 1.93; CI = 1.01; 3.70, *p* = 0.048) and were more likely to be cannabis users (OR = 1.37; CI = 1.02; 1.84, *p* = *0*.04). No differential dropout between the study groups was observed.

### 3.2. Program Use

The initial feedback dialogue, highlighting areas where a participant had adequate resources and those where coaching or counselling was required, was retrieved by 627/688 (91.1%) participants of the intervention group. The most frequently provided coaching topic was stress (67.6%), followed by social media & gaming (48.7%), social skills (25.6%), tobacco/e-cigarette smoking (25.3%), alcohol (20.9%), and cannabis (11.9%).

During the 4 months intervention period, participants were invited via weekly push notification to participate in 16 coaching dialogues (8 for each of the two selected topics). The mean number of completed weekly dialogues among the 688 participants of the intervention group was 2.1 (*SD* = 3.5). Four of ten participants (*n* = 280/688, 40.7%) did not complete any dialogue, another four of ten (271/688, 39.4%) completed one or two dialogues. One of ten participants (*n* = 67/688, 9.7%) completed between 3 and 6 dialogues and another one of ten participants (70/688, 10.2%) completed between 7 and 16 dialogues.

Valid follow-up data concerning program evaluation were available from 378 (54.9%) of the 688 intervention group participants. Of these, 93.4% (*n* = 353/378) indicated that the language and content of the program were comprehensible, 83.1% (*n* = 314/378) reported that the tips and information provided were helpful and 81.2% (309/378) indicated that they perceived the contents as individually tailored to them. Participants in the intervention group who were more engaged, i.e., completed more weekly dialogues, were significantly less likely to drop out compared to the less engaged participants (OR = 0.86; CI = 0.81; 0.92; *p* < 0.01). However, we did not find an association of engagement with the extent of risk reduction based on the primary outcome, suggesting that there was no dose–response effect for our intervention group.

### 3.3. Efficacy of the Intervention Program

The results concerning the primary outcome showed a stronger decrease of addictive behaviors between baseline and follow up in the intervention group compared to the control group (OR_ITT_ = 0.77, CI = 0.67; 0.88, *p* < 0.01; OR_CC_ = 0.71, CI = 0.61; 0.83, *p* < 0.01). Indicating that the odds of an additional risk behavior at follow-up in the intervention group is 0.77 times the odds of an additional risk behavior in the control group. Regarding the secondary outcomes, significant group effects were observed for the following dichotomous measures: at-risk drinking in the preceding 30 days (OR_ITT_ = 0.68, CI = 0.52; 0.89; OR_CC_ = 0.60, CI = 0.43; 0.84), 30 days point prevalence for tobacco/e-cigarette smoking (OR_CC_ = 0.62, CI = 0.40; 0.96) but not for the ITT analysis, and problematic Internet use (OR_ITT_ = 0.61, CI = 0.46; 0.81; OR_CC_ = 0.56, CI = 0.40; 0.79). Furthermore, significant group effects were observed for the following continuous measures: Total number of alcoholic drinks consumed in the preceding 30 days (Cohen’s d_ITT_ = 0.07, d_CC_ = 0.11), number of cannabis use days in the preceding 30 days (Cohen’s d_CC_ = 0.14) but not for the ITT analysis, self-perceived stress (Cohen’s d_ITT_ = 0.27; d_CC_ = 0.18), and extent of problematic Internet use (Cohen’s d_ITT_ = 0.27; d_CC_ = 0.25). The results of the complete-cases (CC) and intention-to-treat (ITT) analyses concerning the dichotomous and continuous outcomes are summarized in Table 2 and Table 3. No significant moderators were identified with our moderation analyses.

## 4. Discussion

### 4.1. Principal Results

The present study tested the efficacy of a mobile app-based coaching program for addiction prevention among apprentices recruited in vocational schools in Switzerland. Three main findings were revealed: (1) Six of ten vocational school students (59%) participated in the study, showing a relatively high interest in the prevention program, (2) program use was relatively low in relation to intended use (3) the program was effective in reducing addictive behaviors, particularly at-risk drinking and problematic Internet use.

Personal recruitment in vocational school classes, in combination with offering a mobile app-based coaching, allowed reaching six of ten apprentices for participation in the *ready4life* program and the associated study. This participation rate of 59% is comparatively high given the program’s 16-week duration and the requirement that users download a separate app on their smartphone. A recent study on the [33] Health4Life app, an intervention to prevent six key risk behaviors among secondary school students in Australia revealed that of the 3610 students provided access to the app, 407 (11%) accessed it. Mobile phone-based programs to reduce problem drinking or to support smoking cessation, conducted in Swiss and German vocational and upper secondary schools, involving similar recruitment procedures, achieved comparable participation rates between 50% and 75% [13,34,35,36,37,38].

Concerning program use, the results show that with an average of 2.1 out of 16 completed weekly dialogues, the average program use was relatively low. However, the low engagement rates are in line with the findings of other reviews on digital interventions to promote mental health [39,40] or to prevent substance use [41,42] in young people that also point at the relatively low levels of user engagement. In the case of the previously mentioned Health4life app, students who accessed the app used it for an average duration of 9.7 days [33]. Nevertheless, increasing program use should be a priority for future optimization. A recent review on factors influencing adherence to mhealth apps revealed that personalization or tailoring of the content to the individual needs of the user offer good starting points for increasing program engagement [41]. This is also in line with previous findings on mobile phone-based interventions demonstrating that tailoring on demographics and psychosocial variables increases intervention efficacy [6]. For the *ready4life* program, a promising strategy could be (1) to assess the needs and optimization suggestions of subsamples of vocational school students with different program engagement rates and similar socially stratifying factors (e.g., male students with migration background and low program use). (2) To involve students of the respective subgroups in the optimization of the program and (3) to implement a program version with advanced tailoring for the corresponding subgroups.

Regarding program efficacy, both the ITT and CC results showed a stronger decrease of addictive behaviors between baseline and follow up in the intervention than the control group. Looking at the secondary outcomes, significant intervention effects were revealed for at-risk drinking and the related quantity of alcohol use, problematic Internet use and perceived stress, while no significant effects were found for cannabis use and tobacco/e-cigarette smoking. Regarding tobacco/e-cigarette smoking prevalence, the CC analysis showed a significant intervention effect, which, however, was narrowly missed in the ITT analysis. Although the effect sizes achieved were small, the study was able to show, in contrast to the findings reported by Champion et al. [8], that a school-based digital intervention to prevent multiple risk behaviors can also be effective in reducing addictive behaviors like problematic Internet and alcohol use.

### 4.2. Limitations

Main limitations of the current study are: (1) since all data are self-reported, there is a chance that social desirability and a probable recall bias may have affected the findings. (2) An examination of the reliability and validity of the composite score used is pending. (3) The study was conducted in 2021/22 during the corona pandemic with varying restrictions depending on the time and place of recruitment. These restrictions also influenced the social life and substance use of young people and accordingly the generalizability of the results of this study. (4) Since we recruited a convenience sample of school classes that were open to participating in the study, the results could not be generalized to vocational school students in Switzerland.

For future efficacy studies, it would be useful to validate the substance use behavior biochemically and to take into account other relevant indicators such as sick days or discontinuation of vocational training.

## 5. Conclusions

*ready4life* is the first mobile app-based coaching program for addiction prevention among apprentices that was tested within a controlled trial. The results of this study suggest that this universally applicable yet individualized intervention approach is effective in reducing risk behaviors in a group of adolescents with a particular high risk of addictive behaviors.

The program could be easily and relatively cheaply implemented by teachers or experts in addiction prevention during one school lesson. Proactive, personal recruitment in vocational school classes, allows reaching the majority of the apprentices present for participation in the *ready4life* program. Increasing program use should be a priority for future optimization and could further increase the effectiveness of the program.

## Figures and Tables

**Figure 1 ijerph-19-15730-f001:**
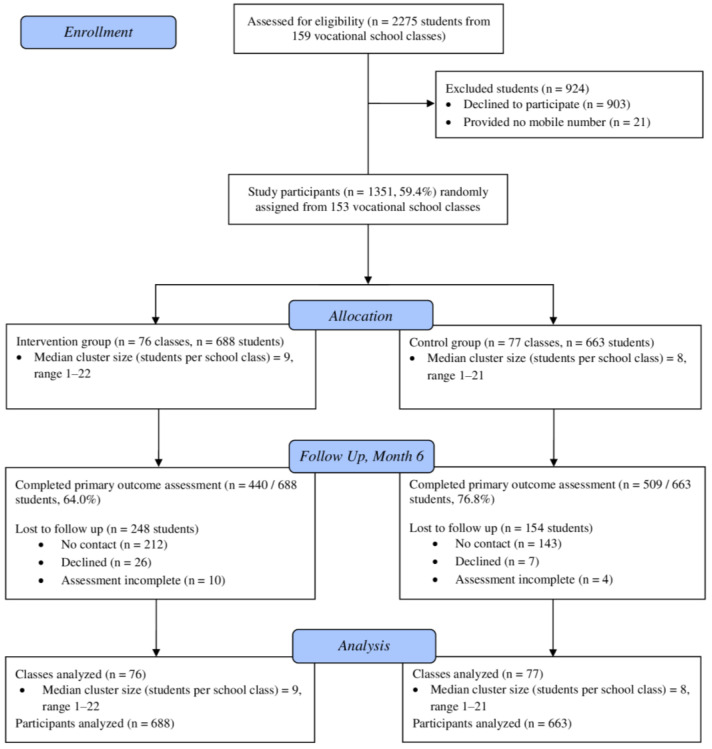
Flow of study participants.

**Table 1 ijerph-19-15730-t001:** Baseline characteristics of the study sample. Values represent *n* (%), unless stated otherwise.

Variable	Intervention *n* = 688	Control *n* = 663	Total *n* = 1351	*p* ^a^
Sex				0.21 ^b^
Male	401 (58.3%)	363 (54.8%)	764 (56.6%)	
Female	287 (41.7%)	300 (45.2%)	587 (43.4%)	
Age, *M* (*SD*)	17.3 (2.7)	17.4 (3.2)	17.3 (3.0)	0.27 ^c^
Composite measure of risk behaviors, *M* (*SD*)	1.5 (1.2)	1.5 (1.2)	1.5 (1.2)	0.74 ^b^
0 risks	176 (25.6%)	155 (23.4%)	331 (24.5%)	
1 risk	226 (32.8%)	235 (35.4%)	461 (34.1%)	
2 risks	135 (19.6%)	138 (20.8%)	273 (20.2%)	
3 risks	101 (14.7%)	89 (13.4%)	190 (14.1%)	
4 risks	50 (7.3%)	46 (6.9%)	96 (7.1%)	
At risk-drinking in the preceding 30 days				0.12 ^b^
No	475 (69.0%)	430 (64.9%)	905 (67.0%)	
Yes	213 (31.0%)	233 (35.1%)	446 (33.0%)	
Total number of alcoholic drinks consumed in the preceding 30 days, *M* (*SD*)	14.8 (34.5)	15.1 (33.0)	14.9 (33.7)	0.86 ^c^
Tobacco/e-cigarette smoking, preceding 30 days				0.59 ^b^
No	393 (57.1%)	368 (55.5%)	761 (58.3%)	
Yes	252 (36.6%)	246 (37.1%)	498 (36.9%)	
Quantity of cigarettes smoked, preceding 30 days, *M (SD)*	89.9 (199.9)	78.0 (176.0)	84.1 (188.5)	0.25 ^c^
Cannabis use, preceding 30 days				0.19 ^b^
No	521 (75.7%)	523 (78.9%)	1044 (77.3%)	
Yes	167 (24.3%)	140 (21.1%)	307 (22.7%)	
Cannabis use days, preceding 30 days, *M* (*SD*)	2.5 (6.9)	2.1 (6.2)	2.3 (6.6)	0.27 ^c^
Problematic Internet use (CIUS-5) ^d^				0.59 ^b^
No	321 (46.7%)	320 (48.3%)	641 (47.4%)	
Yes	367 (53.3%)	343 (51.7%)	710 (52.6%)	
CIUS-5 score, range 0–20, *M* (*SD*)	8.9 (4.2)	9.0 (4.0)	8.9 (4.1)	0.90 ^c^
General self-efficacy, range 1–5, *M* (*SD*)	3.7 (0.7)	3.8 (0.7)	3.7 (0.7)	0.46 ^c^
Self-perceived stress, range 1–5, *M* (*SD*)	3.1 (1.1)	3.2 (1.1)	3.2 (1.1)	0.12 ^c^

^a^ *p* values for the comparison of the intervention and control group. ^b^ *χ^2^* test. ^c^ *t* test. ^d^ CIUS-5: Short Compulsive Internet Use Scale.

**Table 2 ijerph-19-15730-t002:** Intervention effects for dichotomous outcomes. Values represent *n* (% within sub-sample), unless stated otherwise.

	Intervention Group			Control Group						
	Baseline	Follow-Up	Diff.%	Baseline	Follow-Up	Diff.%	*Coeff*	*p*	OR	95% CI
**Complete-cases analysis ^a^**	*n* = 440	*n* = 440		*n* = 509	*n* = 509					
At-risk drinking past 30 days	140 (31.8%)	83 (18.9%)	−12.9	176 (34.6)	139 (27.3%)	−7.3	−0.51	<0.01	0.60	0.43; 0.84
Tobacco/e-cigarette use past 30 days	153 (34.8%)	119 (27.0%)	−7.8	178 (35.0%)	166 (32.6%)	−2.4	−0.48	0.03	0.62	0.40; 0.96
Cannabis use past 30 days	98 (22.3%)	82 (18.6%)	−3.7	99 (19.4%)	82 (16.1%)	−3.3	0.10	0.63	1.11	0.73; 1.70
Problematic Internet use	237 (53.9%)	144 (32.7%)	20.7	265 (52.1%)	219 (43.0%)	−9.1	−0.58	<0.01	0.56	0.40; 0.79
**Intention-to-treat analysis ^a^**	*n* = 688	*n* = 688		*n* = 663	*n* = 663					
At-risk drinking past 30 days	213 (31.0%)	138 (20.1%)	−10.9	233 (35.1%)	182 (27.5%)	−7.6	−0.38	<0.01	0.68	0.52; 0.89
Tobacco/e-cigarette use past 30 days	252 (36.6)	179 (26.0%)	−10.6	246 (37.1%)	209 (31.5%)	−5.6	−0.30	0.06	0.74	0.55; 1.01
Cannabis use past 30 days	167 (24.3%)	128 (18.6%)	−5.7	140 (21.1%)	99 (14.9%)	−6.2	0.26	0.16	1.29	0.90; 1.85
Problematic Internet use	367 (53.3%)	222 (32.3%)	−21.0	343 (51.7%)	280 (42.2%)	−9.5	−0.49	<0.01	0.61	0.46; 0.81

^a^ Generalized mixed models with a random effect for school classes, group as a fixed factor, follow-up scores as outcomes and baseline scores as covariates.

**Table 3 ijerph-19-15730-t003:** Intervention effects for continuous outcomes.

	Intervention Group			Control Group					
	Baseline	Follow-up	Diff.	Baseline	Follow-up	Diff.	*Coeff*	*p*	*d* ^a^
**Complete-cases analysis ^b^**	*n* = 440	*n* = 440		*n* = 509	*n* = 509				
Quantity of alcohol use past 30 days, *M* (*SD*)	14.5 (31.8)	7.0 (15.7)	−7.5	13.9 (31.2)	9.8 (17.4)	−4.1	−2.81	0.01	0.11
Quantity of cigarettes smoked past 30 days, *M* (*SD*)	81.4 (195.5)	55.3 (155.4)	−26.1	71.7 (167.8)	57.9 (148.7)	−13.8	−8.58	0.20	0.07
Cannabis smoking days past 30 days, *M* (*SD*)	2.5 (7.0)	1.5 (5.3)	−1.0	1.8 (5.7)	1.7 (6.1)	−0.1	−0.61	0.03	0.14
Perceived stress past 30 days, *M* (*SD*)	3.1 (1.1)	2.7 (1.1)	−0.4	3.2 (1.1)	3.0 (1.1)	−0.2	−0.26	<0.01	0.18
General self-efficacy, *M* (*SD*)	3.7 (0.7)	3.8 (0.7)	−0.1	3.8 (0.7)	3.7 (0.7)	−0.1	0.07	0.16	−0.29
Problematic Internet use, *M* (*SD*)	8.9 (4.2)	6.8 (4.1)	−2.1	8.9 (3.9)	7.8 (4.0)	−1.1	−1.03	<0.01	0.25
**Intention-to-treat-analysis ^b^**	*n* = 688	*n* = 688		*n* = 663	*n* = 663				
Quantity of alcohol use past 30 days, *M* (*SD*)	14.8 (34.5)	6.8 (15.3)	−8.0	15.1 (33.0)	9.5 (16.9)	−5.6	−2.66	<0.01	0.07
Quantity of cigarettes smoked past 30 days, *M* (*SD*)	89.9 (199.9)	54.2 (155.4)	−35.7	78.0 (176.0)	55.5 (144.9)	−22.5	−6.0	0.40	0.07
Cannabis use days past 30 days, *M* (*SD*)	2.5 (6.8)	1.7 (5.7)	−0.8	2.1 (6.3)	1.7 (6.0)	−0.4	−0.07	0.81	0.06
Perceived stress past 30 days, *M* (*SD*)	3.1 (1.1)	2.7 (1.1)	−0.4	3.2 (1.1)	3.1 (1.1)	−0.1	−0.28	<0.01	0.27
General self-efficacy, *M* (*SD*)	3.7 (0.7)	3.8 (0.8)	0.1	3.8 (0.7)	3.8 (0.7)	0	0.05	0.24	−0.14
Problematic Internet use, *M* (*SD*)	8.9 (4.2)	6.8 (4.0)	−2.1	9.0 (4.0)	8.0 (4.0)	−1	−1.20	<0.01	0.27

^a^ Effect sizes Cohen’s *d* were calculated based on baseline-follow-up differences between the two study groups [32]. ^b^ Linear mixed models with a random effect for school classes, group as a fixed factor, follow-up scores as outcomes and baseline scores as covariates.

## Data Availability

Data available on request due to restrictions. The datasets generated and analyzed during the current study are not publicly available due to the Swiss data protection law but are available from the corresponding author on reasonable request. Requests will be reviewed for reasonability and compliance with the study purpose and the participants’ informed consent.

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
