# Peer review of "Efficacy of a Mobile App-Based Coaching Program for Addiction Prevention among Apprentices: A Cluster-Randomized Controlled Trial"

_ijerph, 2022, doi:10.3390/ijerph192315730_

Round 1

Reviewer 1 Report

It is not clear to me why a non-parametric test, the Wilcoxon-Mann-Whitney-test, has been employed for sample size calculation whereas all subsequent analyses were done at the parametric level.

 The primary outcome was a composite measure of four very different risk behaviors. This may be quite heterogeneous (apples and oranges), so I wonder what the internal consistency of this composite measure is, and whether there is a theoretical rationale to collapse such different variables.

 If intervention engagement is a predictor of missing data at follow-up, why is engagement not associated with outcome levels?

Reviewer 2 Report

When I reviewed the submitted manuscript, I evaluated it as a valuable study to verify the effectiveness of the app-based intervention for addiction prevention. However, I think there should be a lot of supplementation and revision in writing the manuscript. These are the things that need to be supplemented:

1. Abstract: The description of the participant must be described in the Methods part. When summarizing results in abstract, you should avoid statistics.

2. Introduction: 1) App-based interventions, especially app-based addiction treatments, have been applied a lot, and their effects have been verified. The reason why this study is necessary or rationale needs to be clearly presented in the introduction. 2) The short paragraphs of the introduction are legible or good for readability, but there is no depth in explaining.

3. Objectives and research design are not appropriate for Materials and Methods. It must be at the end of the introduction.

4. If this is not the first time to study the effectiveness of app-based addiction prevention treatment, what is important in this study is the contents of the app intervention program. That is why the evidences should have been presented for each content of the intervention program for adolescents.

5. Discussion: 1) The clinical implications of the research results are not properly presented in the discussion section. Based on the results found in this study, please describe in detail how app-based addiction prevention should be performed. 2) Overall, the content of the discussion should be enriched discussion.

6. Limitation: While describing the limitations of this study, it is also necessary to suggest how to overcome these limitations in future studies.

Round 2

Reviewer 2 Report

Thank you for revising the manuscript based on comments in the first round of review. .